# Atezolizumab plus stereotactic ablative radiotherapy for medically inoperable patients with early-stage non-small cell lung cancer: a multi-institutional phase I trial

Arta M. Monjazeb [1,7,8], Megan E. Daly [1,7,8] ✉, Guillaume Luxardi [1], Emanual Maverakis[1], Alexander A. Merleev[1], Alina I. Marusina[1], Alexander Borowsky [1], Amin Mirhadi[2], Stephen L. Shiao[2], Laurel Beckett[1], Shuai Chen [1], David Eastham[3], Tianhong Li [1], Logan V. Vick[1], Heather M. McGee [4], Frances Lara[1], Leslie Garcia[1], Leigh Anne Morris[1], Robert J. Canter [1], Jonathan W. Riess[1], Kurt A. Schalper [5], William J. Murphy [1] & Karen Kelly[1,6,8]

Stereotactic ablative radiotherapy (SABR) is a standard-of-care for medically-inoperable-early-stage non-small cell lung cancer (NSCLC). One third of patients progress and chemotherapy is rarely used in this population. We questioned if addition of the immune-checkpoint-inhibitor (ICI) atezolizumab to standard-of-care SABR can improve outcomes. We initiated a multi-institutional single-arm phase I study (NCT02599454) enrolling twenty patients with the primary endpoint of maximum tolerated dose (MTD); secondary endpoints of safety and efficacy; and exploratory mechanistic correlatives. Treatment is well tolerated and full dose atezolizumab (1200 mg) is the MTD. Efficacy signals include early responses (after 2 cycles of ICI, before initiation of SABR) in 17% of patients. Biomarkers of functional adaptive immunity, including T cell activation in the tumor and response to ex-vivo stimulation by circulating T cells, are highly predictive of benefit. These results require validation and are being tested in a phase III randomized trial.

Immune checkpoint inhibitors (ICI) targeting programmed death (PD)-1 and programmed death ligand (PD-L1) have become a component of standard front-line therapy for metastatic[1,2], locally advanced[3,4], and resectable[5,6] non-small cell lung cancer (NSCLC). Several mechanistic biomarkers for ICI efficacy have been reported, including PD-L1 expression[7], density and dysfunction of T cells in the TME[8], and tumor mutational burden[9].

Immunotherapy may also be very effective in eradicating micro-metastatic disease in early-stage patients, given its proven efficacy in large, bulky tumors that have established immune suppression systemically and within the tumor microenvironment (TME). Clinical data supports this hypothesis with the randomized phase III trials IMpower 010 and CheckMate 816 demonstrating efficacy for adjuvant or neoadjuvant administration of ICIs leading to FDA approvals[5,6].

[1]UC Davis Health, Sacramento, CA 95817, USA. [2]Cedars-Sinai Medical Center, Los Angeles, CA 90048, USA. [3]David Grant USAF Medical Center, Travis AFB, Fairfield, CA 93405, USA. [4]City of Hope Cancer Center, Duarte, CA 91010, USA. [5]Yale School of Medicine, New Haven, CT 06519, USA. [6]International Association for the Study of Lung Cancer, Denver, CO 80202, USA. [7]These authors contributed equally: Arta M. Monjazeb, Megan E. Daly. [8]These authors jointly supervised this work: Arta M. Monjazeb, Megan E. Daly, Karen Kelly. ✉e-mail: medaly@ucdavis.edu

For medically inoperable early-stage NSCLC patients who cannot tolerate surgical resection or chemotherapy due to comorbid conditions, stereotactic ablative radiotherapy (SABR) is a standard of care. Clinical studies demonstrate rates of in-field tumor control exceeding 90% at 3 years[10,11]. However, rates of regional and distant failure have remained unacceptably high, with distant recurrence rates exceeding 30% at 5 years with long-term follow up[12,13]. Recurrence rates are even higher in patients with high-risk features, including large tumor diameter[10,14,15], high standardized uptake value (SUV) on fluorodeoxyglucose (FDG) positron emission tomography (PET)[16], and high histologic grade[17], among others. In a multi-institutional study evaluating SABR for tumors >5 cm, disease-free survival (DFS) at 2 years was only 53.5%[18], and increasing SUV was an independent predictive factor for decreased survival. In a National Cancer Institute Database analysis, overall survival (OS) at 3 years was only 33% among patients with tumors >4 cm treated with SABR alone[19]. For surgically managed early-stage NSCLC, adjuvant chemotherapy is frequently offered for node-negative tumors larger than 4 cm based on prospective randomized trials suggesting a benefit for select patients[20–22]. However, adjuvant chemotherapy has not been consistently offered to SABR patients due to a lack of supporting prospective data and a lack of tolerance to platinum-

based chemotherapy in this typically frail population. ICI's milder side effect profile is ideal for evaluation in the medically inoperable population. Furthermore, preclinical[23] and early clinical[24] data suggest potential synergy between radiotherapy and ICI, although the optimal timing of radiation plus ICI is not well defined.

We conducted a multi-institutional phase I study with an expansion cohort testing the addition of six cycles of atezolizumab to SABR in high-risk, medically inoperable, early-stage NSCLC accompanied by in-depth correlative analyses (Fig. 1A). Dose-limiting toxicity (DLT) and safety analyses found escalation to full dose atezolizumab (1200 mg) is well tolerated in combination with SABR. Preliminary signals of ICI efficacy in this population were also observed with early responses to ICI after just two cycles and before initiation of SABR, and a dose–response effect on post-hoc analysis with significantly improved freedom from progression (FFP) in patients receiving full dose atezolizumab compared to dose levels 1 and 2. The correlative analysis demonstrated that an immune signature of heightened T cell functionality in the TME and periphery correlates with clinical outcomes, as opposed to biomarkers such as T cell dysfunction or dormancy reported in advanced stage NSCLC, suggesting biological and mechanistic differences in the immune landscape of early vs. advanced stage NSCLC. Conclusions regarding

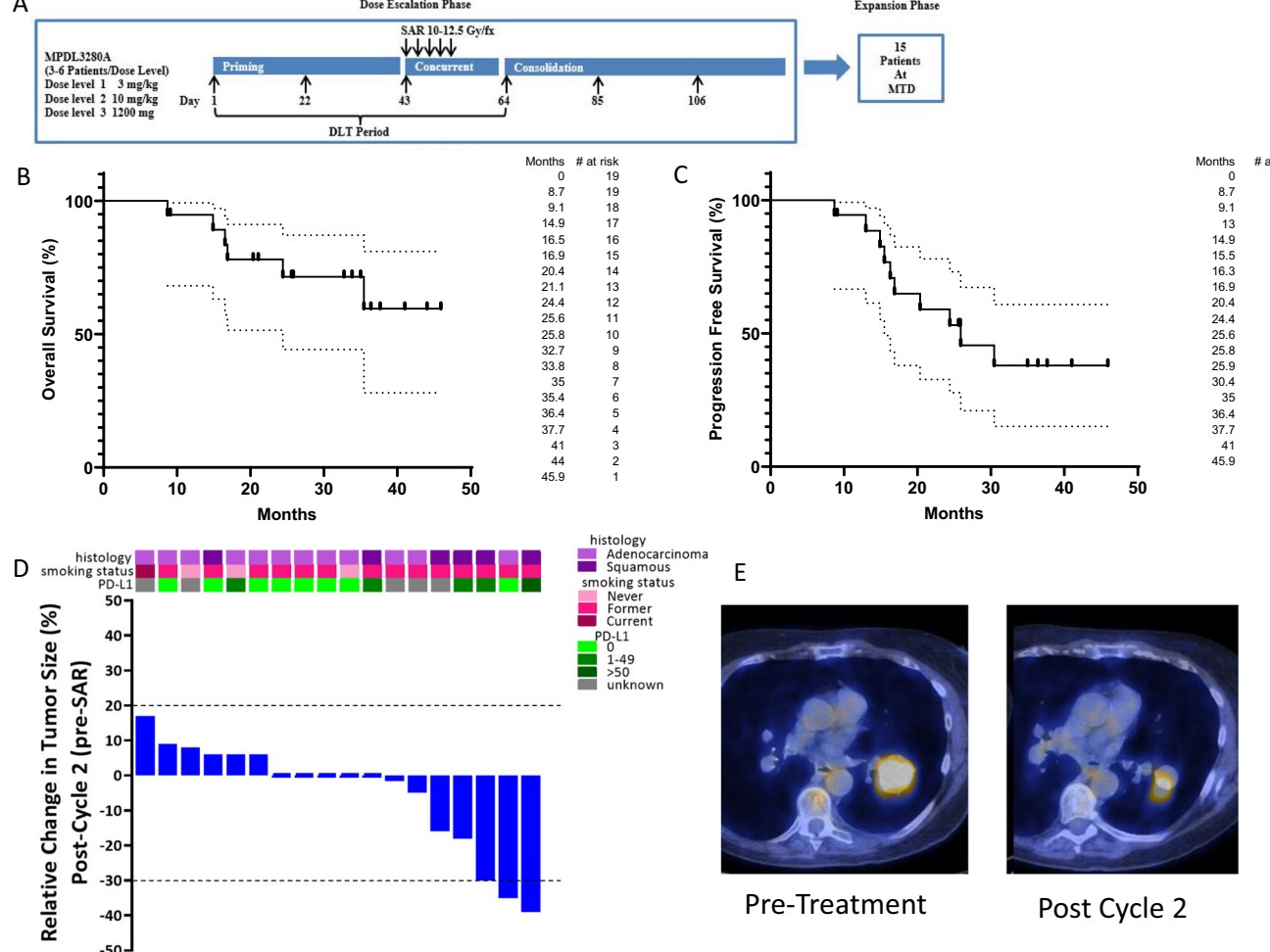

**Fig. 1 | Clinical outcomes. A** Schema of a clinical trial. **B** Overall survival of clinical trial patients from the date of enrollment estimated by the Kaplan–Meier method (n = 19). **C** Progression-free survival of clinical trial patients from the date of enrollment by the Kaplan–Meier method (n = 18). **D** Waterfall plot depicting the relative change in tumor volume following two cycles of neoadjuvant atezolizumab stratified by PD-L1 status, smoking status, and histology (n = 18). The bars at the top

of the plot represent smoking status, histology, and PD-L1 tumor proportion score for each patient. **E** Axial fused PET/CT images from a patient with squamous cell carcinoma with marked response following two cycles induction atezolizumab. Pre-treatment tumor measuring 4.1 cm with $SUV_{max}$ 24.9 is shown in the left panel, and a post-treatment tumor measuring 2.5 cm with $SUV_{max}$ 17.2 is shown in the right panel. Source data are provided as a Source Data file.

## Table 1 | Patient characteristics

| Age (median) | 76.0 (62.2–88.9) |
|---|---|
| *Gender* | |
| Male | 9 (45%) |
| Female | 11 (55%) |
| *Smoking status* | |
| Current | 1 (5%) |
| Former | 16 (80%) |
| Never | 3 (15%) |
| *Zubrod performance status* | |
| 0 | 8 (40%) |
| 1 | 9 (45%) |
| 2 | 3 (15%) |
| *Pulmonary function* | |
| FEV1 (L) Median (range) | 1.58 (0.84–2.61) |
| FEV1 (%) Median (range) | 61.5 (36–130) |
| *T stage (AJCC 8th edition)* | |
| T1a | 0 |
| T1b | 9 (45%) |
| T1c | 3 (15%) |
| T2a | 6 (30%) |
| T2b | 2 (10%) |
| *Histology* | |
| Adenocarcinoma | 13 (65%) |
| Squamous Cell | 7 (35%) |
| Tumor Diameter Median (range) | 2.4 cm (1.1–4.4 cm) |
| Pre-treatment $SUV_{max}$ Median (range) | 5.8 (2.0–24.9) |
| *High-risk features* | |
| Diameter > 1 cm | 20 (100%) |
| $SUV_{max}$ > 6.3 | 7 (35%) |
| Moderately/poorly differentiated | 9 (45%) |
| *Number of high-risk features* | |
| 1 | 7 (35%) |
| 2 | 10 (50%) |
| 3 | 3 (15%) |

## Table 2 | Treatment-related adverse events. Listed as the number of patients experiencing each adverse event

| Treatment-related adverse event | Grade 1 | Grade 2 | Grade 3 |
|---|---|---|---|
| Anemia | 8 | 2 | 0 |
| Hyperthyroidism | 2 | 1 | 0 |
| Hypothyroidism | 1 | 2 | 0 |
| Diarrhea | 3 | 0 | 0 |
| Nausea | 2 | 2 | 0 |
| Edema | 1 | 1 | 0 |
| Fatigue | 7 | 2 | 0 |
| Chest Pain | 0 | 1 | 0 |
| LFT abnormality | 3 | 0 | 1 |
| Lymphopenia | 3 | 6 | 3 |
| Neutropenia | 1 | 0 | 0 |
| Bronchial infection | 0 | 1 | 0 |
| Thrombocytopenia | 3 | 0 | 0 |
| Decreased WBC | 6 | 0 | 0 |
| Anorexia | 1 | 1 | 0 |
| Electrolyte abnormalities | 2 | 0 | 0 |
| Dizziness | 2 | 0 | 0 |
| Headache | 1 | 0 | 0 |
| Peripheral neuropathy | 1 | 0 | 0 |
| Dyspnea | 2 | 1 | 0 |
| Urinary Frequency | 1 | 0 | 0 |
| Pneumonitis | 1 | 1 | 0 |
| Cough | 0 | 2 | 0 |
| Rash | 3 | 1 | 1 |
| Musculoskeletal | 2 | 0 | 2 |
| Conjunctivitis | 0 | 1 | 0 |
| Ear disorder | 0 | 1 | 0 |

efficacy and correlative outcomes are limited by small patient numbers and the early-stage nature of this trial and should be viewed as hypothesis-generating.

## Results

### Patient enrollment and characteristics

From April 2016–Aug 2019, twenty patients enrolled, including 15 on the dose-finding component and 5 in the expansion cohort. Three patients in the dose-finding phase discontinued treatment early for non-DLT reasons (travel hardship, chronic obstructive pulmonary disease exacerbation, and grade 2 liver function testing (LFT) abnormality) and were replaced. Overall, four patients (1 non-evaluable for DLT or response) were treated at dose level 1, 6 at dose level 2, 5 at dose level 3 (2 non-evaluable for DLT, 1 non-evaluable for response), and an additional 5 patients were treated at dose level 3 in the expansion cohort.

Patients included 9 men (45%) and 11 women (55%), with a median age of 76.0 years (range: 62.2–88.9 years). Three patients (15%) had a Zubrod PS of 2, and 17 (85%) had a PS of 0–1. Sixty-five percent of enrolled patients had at least 2 of the three pre-defined high-risk features [size ≥ 2 cm, $SUV_{max}$ ≥ 6.2, and grade ≥ 2]. Among 13 evaluable patients with sufficient baseline/archival tumor tissue for testing, PD-L1 tumor proportion score (TPS) was 0% for 8 (62%), >1–50% for 4

(31%), and >50% for 1 (8%) (Supplemental Fig. 1, Fig. 1D). Full patient and tumor characteristics are shown in Table 1.

### Treatment course and toxicity

Among 15 patients on the phase 1 component, 12 were evaluable for DLT. Four patients enrolled on dose level 1 (3 mg/kg). One patient withdrew from the trial due to travel hardship and was replaced. The remaining patients had no DLTs and completed six cycles of atezolizumab. Six patients enrolled on dose level 2 (10 mg/kg). One of the first three patients developed a grade 3 rash and was removed from protocol therapy for DLT but continued study evaluation. An additional three patients enrolled at dose level 2 and completed six cycles of protocol therapy without DLT. Five patients enrolled on dose level 3 (1200 mg). Two patients on dose level 3 discontinued protocol therapy early for non-DLT reasons after cycle 2 (grade 2 LFT abnormalities and a COPD exacerbation) and were replaced. The other 3 patients completed six planned cycles of therapy without DLT, and this was declared the recommended phase 3 dose. Five patients enrolled in the expansion cohort, one of whom discontinued protocol therapy after cycle 5 due to toxicity after experiencing dose delays for cycles 2 and 5. Other grade 3 adverse events not meeting DLT criteria included LFT abnormalities outside the 9-week DLT window (n = 1), lymphopenia (n = 3), and musculoskeletal events (n = 2). Two patients developed pneumonitis (one grade 1 and one grade 2). Grade 3 pneumonitis was not observed. All treatment-related adverse events are shown in Table 2, and all adverse events are shown in Supplemental Table 3.

All patients, including those who discontinued protocol treatment early, received the full planned course of SABR. All SABR plans met pre-specified normal tissue dose constraints. Six patients were

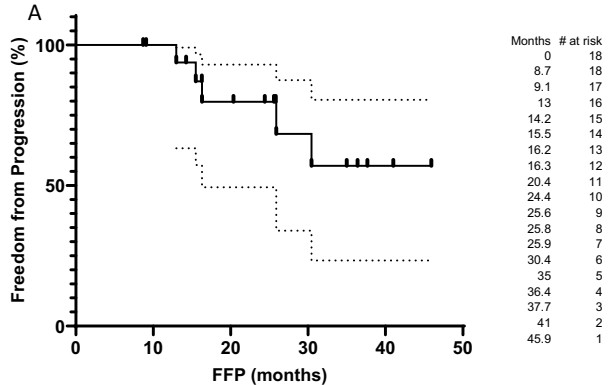

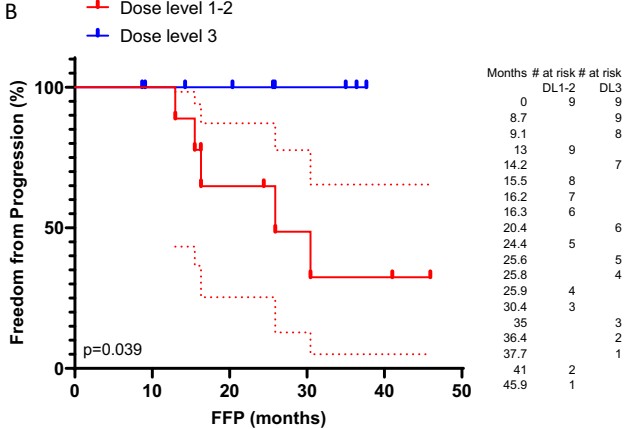

**Fig. 2 | Freedom from progression. A** Freedom from progression for the entire cohort from the date of enrollment by the Kaplan–Meier method ($n = 18$). **B** Freedom from progression stratified by a dose of atezolizumab administered ($n = 18$). Source data are provided as a Source Data file.

treated to 50 Gy over 4 fractions and 13–50 Gy over 5 fractions. One patient who came off protocol prior to SABR received 54 Gy in 3 fractions.

### Treatment outcomes

As outlined above, 18/20 patients were available for response/progression analysis, and 19/20 patients were available for survival analysis. At the time of analysis, 11 patients were alive, and 8 had expired. Of the 8 expired patients, 3 had evidence of disease progression, 1 succumbed to a second primary malignancy, and 4 had no evidence of cancer or therapy-related death. Median follow-up was 25.8 months (range 7.6–41 months), and median OS was not reached (Fig. 1B). At the time of analysis, 5/18 evaluable patients (28%) had progression, and median progression-free survival (PFS) was 25.9 months (Fig. 1C). No patient progressed on protocol treatment prior to initiation of SABR (Fig. 1D). No patients had local (in-field) progression alone. One patient progressed at the ablated lesion (in-field) and with metastatic disease (bone). The other 4 progressions were out of the irradiated field (metastatic–brain, metastatic–contralateral lung, regional–out of field lung and metastatic–liver, and regional–out of field lung).

We evaluated early treatment response to two cycles of atezolizumab (before initiation of SABR). Evaluation of ICI responses after SABR is not possible. Unconfirmed partial responses were observed in 3/18 patients (17%), and three additional patients had a minor response with tumor reduction <30% (Fig. 1D, E). Early responses were observed in 2 of the 5 pts with >1% PD-L1 expression and 1 of the 8 pts with 0% PD-L1 expression (Fig. 1D). Efficacy and response data in this small phase 1 trial should be viewed as hypothesis generating.

### Post-hoc analysis of FFP

FFP was analyzed post-hoc since deaths from intercurrent illness limited the utility of PFS to evaluate progression. Median FFP was not reached (Fig. 2A). We found no significant association between PD-L1 expression (>1% TPS) and FFP (Supplemental Fig. 2A), nor between early response and FFP (Supplemental Fig. 2B).

A dose–response effect was apparent. All early responses occurred at dose level 3. Also, greater than 50% of patients at dose level 1 or 2 progressed, but no patients at dose level 3 progressed. Patients at dose level 3 had a significantly improved FFP compared to those at dose levels 1 and 2 (median: not reached vs. 25.9 months, hazard ratio: 0.156 (0.027-0.911), Fig. 2B). This analysis is hypothesis generating, and the efficacy of this approach is being tested in a randomized phase III trial.

### Tissue analysis

A biomarker of interest for ICI is a T cell inflamed TME[25]. Using a previously validated multi-plex QIF T cell activation panel demonstrated to correlate with ICI response in advanced NSCLC[26], we analyzed baseline tumor samples. In samples from nine patients with sufficient tumor tissue to stain (five FNA, four core biopsies), CD3+ TIL density was predictive of both early response and FFP in these nine patients (Fig. 3A–D). Additional lung biopsies for more tissue from the other patients and on-treatment biopsies were not feasible or safe in this frail population. Patients with early response (Fig. 3A, B) or FFP (Fig. 3C, D) had higher levels of CD3+ TILs. The CD3 QIF score was twofold higher in responders (as assessed after two cycles of atezolizumab and before SABR) compared to non-responders and in non-progressors compared to progressors. Unlike previous studies demonstrating T cell dormancy[26] or dysfunction[27,28] in the TME as a biomarker for ICI response, we found T cell functionality correlated with improved response and outcomes (Fig. 3E–J). Higher levels of T cell proliferation (Ki-67) and effector function (GZB) were significantly associated with early ICI response (Fig. 3E–G) and FFP (Fig. 3H–J). Levels of proliferating and functional T cells were threefold higher in responders and non-progressors (Fig. 3E–J). We found no correlation between PD-L1 expression and TIL predominance or functionality in these samples (Supplemental Fig. 3).

### Peripheral blood analysis

We employed four multi-color flow cytometry panels (Supplemental Table 2) to analyze treatment-induced changes in the proportion and functional status of immune cells and subsets in peripheral blood. Baseline blood samples were available for 17/20 patients. Four cell subsets had statistically significant changes over treatment. We observed significant decreases in circulating PD-L1+ myeloid-derived suppressor cells (MDSCs) and monocytes (Fig. 4A, B) and increases in activation of memory CD8 and CD4 T cells (Fig. 4C, D) as measured by ICOS expression on PD1+ Ki-67+ T cells. Treatment-induced decreases in PD-L1+ suppressive cells and increases in activation of PD1+ T cells are consistent with other reports[29].

Transcriptomic analysis of PBMCs from patients with sufficient samples demonstrated no significant changes in gene expression or clustering across the course of therapy (Supplemental Fig. 4). The top differentially expressed genes did not cluster samples (Supplemental Fig. 4A), and principal components analysis (PCA) did not group gene expression by treatment cycle (Supplemental Fig. 4B). Given markers of T cell function and activation observed in the TME (Fig. 3) and PBMCs (Fig. 4) we examined the BIOCARTA T cell receptor signaling gene set which also failed to differentiate samples by treatment cycle (Supplemental Fig. 4C, D). There were, however, gene expression differences across cycles, and we could retrospectively construct a gene set consisting largely of immune-related and proliferation-related genes, which clustered and grouped samples pre- vs. post-treatment (Supplemental Fig. 5).

We next evaluated differences in PBMCs between progressors and non-progressors. We found statistically significant differences

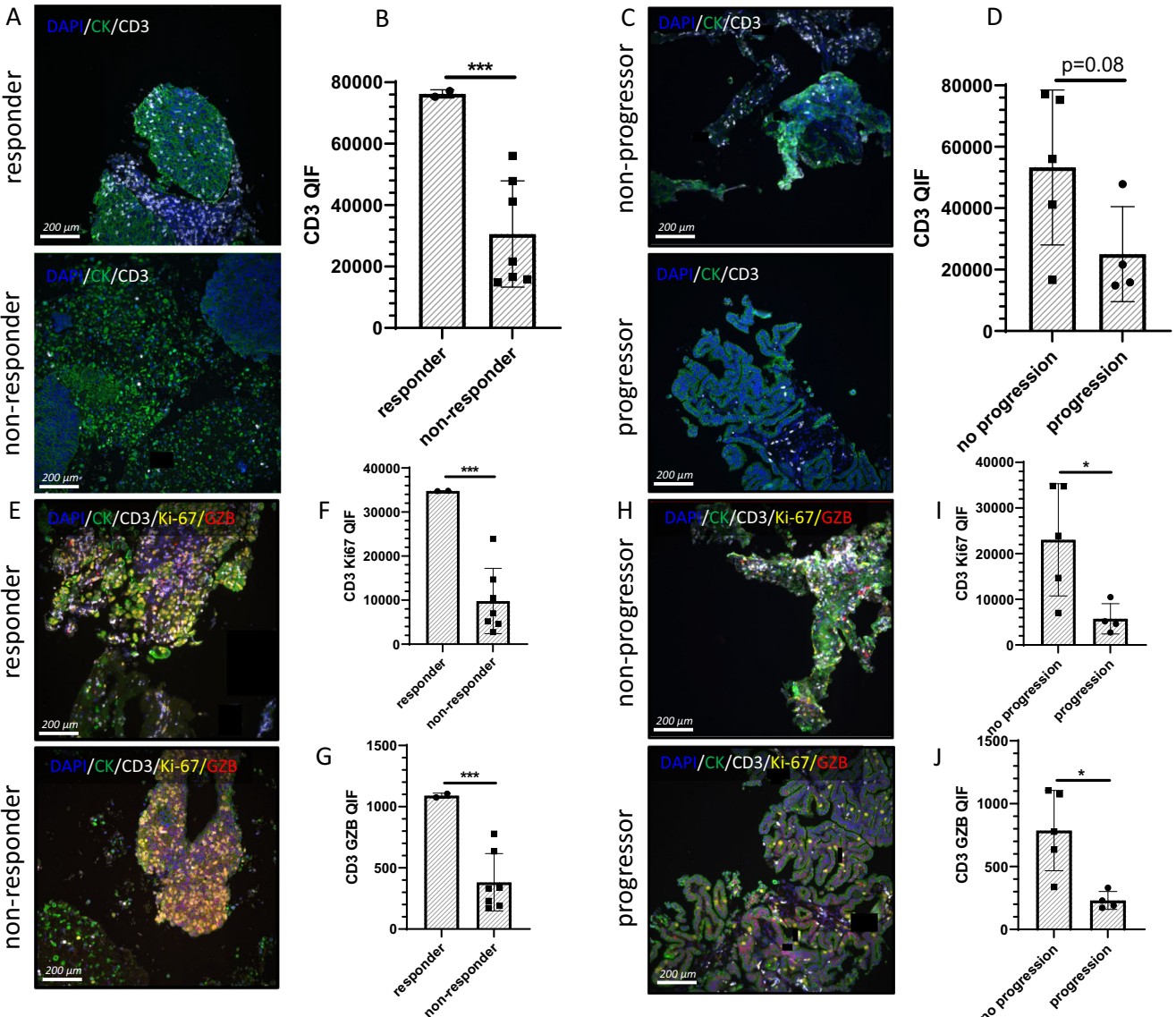

**Fig. 3 | Multiplex immuno-fluorescence of the TME. A–D** Staining of tissue sections for CD3 expression in the TME (*n* = 9). Nine patients with sufficient tissue for analysis were stained. This included two with the early response and seven without, four with progression, and five without. As outlined in the text, patients with early response did not progress, and thus the early response group is a subset of the non-progressor group. CD3 is stained in white, DAPI in blue, and cytokeratin in green. **A** Representative staining for CD3 expression in a patient with early response to atezolizumab (top panel) or no response to atezolizumab (bottom panel). **B** Bar graph representing the mean quantitative immuno-fluorescence of CD3 staining in responders and non-responders (*p* = 0.0004). **C** Representative staining for CD3 expression in a patient free from progression (top panel) or with disease progression (bottom panel). **D** Bar graph representing the mean quantitative immuno-fluorescence of CD3 staining in non-progressors and progressors. **E–J** Multiplex staining for CD3, Ki67, and granzyme b (GZB) expression (*n* = 9). CD3 is stained in white, Ki67 in yellow, GZB in red, DAPI in blue, and cytokeratin in green.

**E** Representative multiplex staining in a patient with early response to atezolizumab (top panel) or no response to atezolizumab (bottom panel). **F** Bar graph representing the mean quantitative immuno-fluorescence of Ki67 staining in CD3+ cells in responders and non-responders (*p* = 0.0001). **G** Bar graph representing the mean quantitative immuno-fluorescence of GZB staining in CD3+ cells in responders and non-responders (*p* = 0.0002). **H** Representative multiplex staining for CD3, Ki67, and GZB expression in a patient free from progression (top panel) or with disease progression (bottom panel). **I** Bar graph representing the mean quantitative immuno-fluorescence of Ki67 staining in CD3+ cells in non-progressors and progressors (*p* = 0.0320). **J** Bar graph representing the mean quantitative immuno-fluorescence of GZB staining in CD3+ cells in non-progressors and progressors (*p* = 0.0159). Statistical comparisons between groups were performed with a two-sided *t*-test. The center line represents the mean, and the error bars represent the standard deviation of the mean. *$p \leq 0.05$; ***$p \leq 0.001$. Source data are provided as a Source Data file.

in several parameters, primarily present at baseline and sustained through therapy (Supplemental Fig. 6A–H). Based on this observation, we hypothesized that pre-treatment baseline biomarkers may identify progressors. We revisited the PBMC transcriptomic analysis. Expression differences in the top differential expressed genes at baseline (Supplemental Fig. 7) failed to separate progressors from non-progressors. The BIOCARTA T cell receptor signaling gene set (Supplemental Figure 8) demonstrated

statistically significant (*p* = 0.016) expression differences, and both cluster analysis and PCA separated progressors from non-progressors (Supplemental Figure 8, Fig. 5A, B). Twelve genes in this gene set, all of which have essential roles in T cell function, had significant differential expression in progressors versus non-progressors (Fig. 5A–C).

Flow cytometry also demonstrated significant baseline differences in the frequency of major T cell subsets between progressors

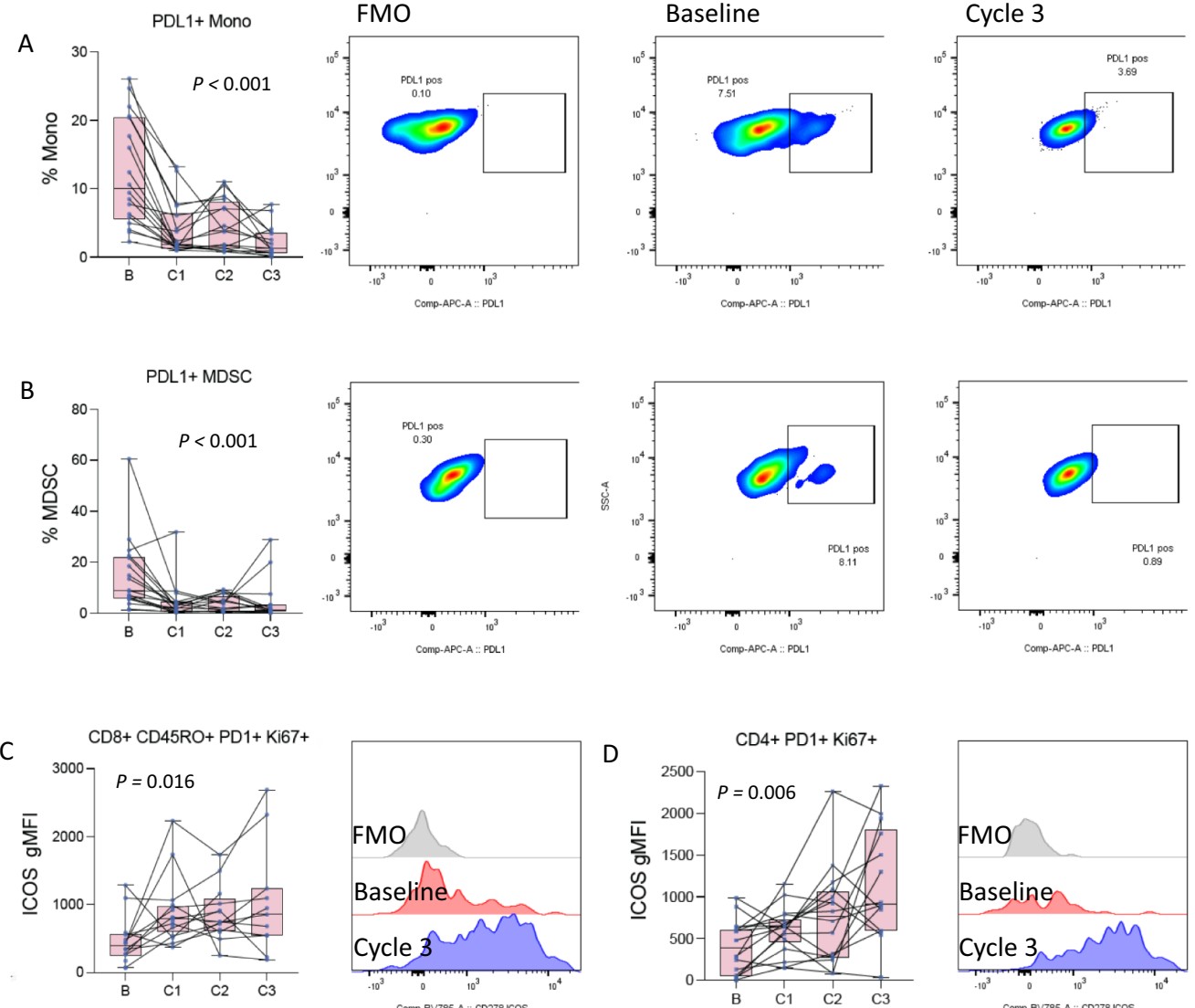

**Fig. 4 | Changes in PBMCs during therapy. A** Changes in the frequency of PDL1 positive monocytes at baseline and after each of the first 3 cycles of atezolizumab ($n = 19, 19, 17, 16$ patients at baseline, C1–C3). The left panel is a box and whisker plot where each dot represents the value for an individual patient, the line represents the median, the box represents the interquartile range, and the whiskers represent the spread of the data. The overlayed line graph demonstrates the trend for the individual patients across cycles. The right panel is representative flow cytometry staining for an individual patient at baseline and after cycle 3, as well as the fluorescence minus one (FMO) negative gating control. **B** Changes in the frequency of PDL1 positive myeloid-derived suppressor cells (MDSC) at baseline and after each of the first 3 cycles of atezolizumab atezolizumab ($n = 19, 19, 17, 16$ patients at baseline, C1–C3). The left panel is a box and whisker plot representing the median, interquartile range, and data spread with an overlayed line graph demonstrating the trend for the individual patients across cycles. **C** Changes in the mean

fluorescence intensity (MFI) of ICOS on PD1/Ki67 double positive memory CD8 T cells at baseline and after each of the first 3 cycles of atezolizumab atezolizumab ($n = 14, 17, 13, 11$ patients at baseline, C1–C3). The left panel is a box and whisker plot representing the median, interquartile range, and data spread with an overlayed line graph demonstrating the trend for the individual patients across cycles. Error bars represent the spread of the data. The right panel is a representative flow cytometry histogram for an individual patient at baseline and after cycle 3, as well as the corresponding FMO negative gating control. **D** Changes in the MFI of ICOS on PD1/Ki67 double positive CD4 T cells at baseline and after each of the first 3 cycles of atezolizumab ($n = 18, 18, 16, 14$ patients at baseline, C1–C3). The left panel is a box and whisker plot representing the median, interquartile range, and data spread with an overlayed line graph demonstrating the trend for the individual patients across cycles. Statistical comparisons across the course of therapy were performed by ANOVA. Source data are provided as a Source Data file.

and non-progressors (Fig. 6, Supplemental Fig. 9). Using receiver operating characteristics (ROC) curves, these parameters accurately classified patients who progressed. Baseline circulating CD8+ T cell frequency trended towards elevated levels in progressors vs. non-progressors and was predictive for FFP (Fig. 6A, B). We observed differences in the frequency of PD1 expression on CD8+ T cells (Fig. 6C–E). Elevated CD8+ PD1+ T cells were associated with worse clinical outcomes in contrast with previous reports where elevated PD1 correlated with improved ICI outcomes[30]. The frequency of Tim3+ CD8+ T cells also differed between progressors and non-progressors

and was highly predictive of FFP (Fig. 6F–H). As opposed to PD1 expression, the frequency of Tim3+ CD8+ T cells was significantly elevated in non-progressors ($p = 0.003$, Fig. 6F, G). We also detected differences between progressors and non-progressors in CD4+ and the sparse population of circulating NKT cells (Supplemental Fig. 9). Mirroring CD8+ T cells, the frequency of baseline circulating NKT cells was significantly higher in progressors ($p = 0.007$, Supplemental Fig. 9J–L) but the frequency of TIM3+ NKT cells was significantly higher in non-progressors and strongly predicted FFP ($p = 0.011$, Supplemental Fig. 9M–O).

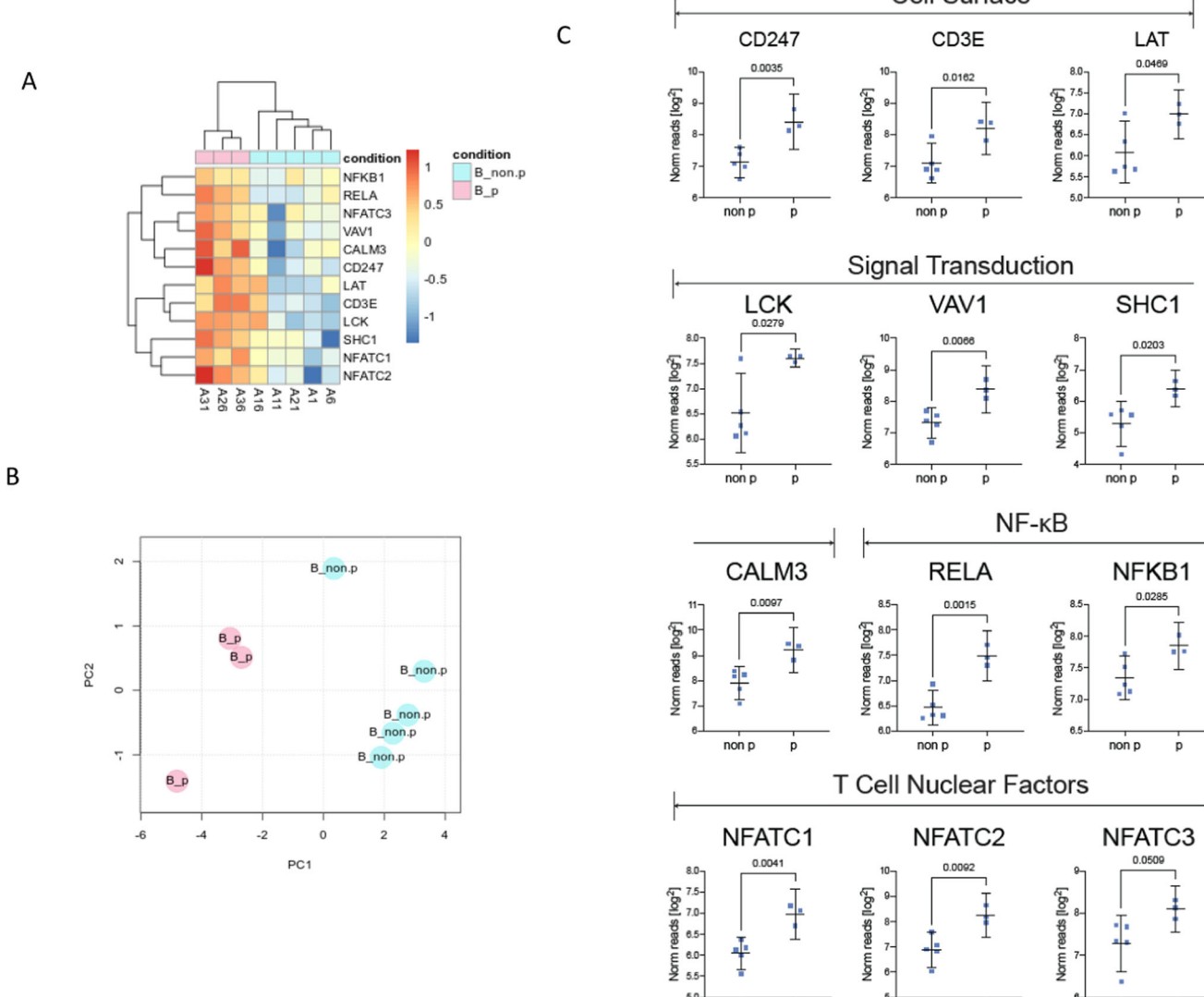

**Fig. 5 | Baseline PBMC transcriptomic differences in progressors versus non-progressors.** Transcriptomic analysis was performed on eight samples with sufficient tissue for analysis (non-progressor, NP: $n = 5$, progressor, P: $n = 3$). **A** Heat map demonstrating expression of genes from the BIOCARTA T cell receptor signaling gene set with clustering analysis separating progressors from non-progressors. **B** Principal components analysis (PCA) demonstrates the separation of progressors from non-progressors based on the expression of 12 genes. **C** Whisker plots depicting the expression of individual genes from the BIOCARTA T cell receptor signaling gene set with significant differential expression at baseline. The whisker plot represents the mean and 95% confidence interval. The points represent individual patient values. *P* values are from a two-sided *t*-test. Source data are provided as a Source Data file.

## Ex-vivo T cell functionality

Transcriptomic and phenotypic analysis of circulating immune cells (Figs. 5 and 6, Supplemental Figs. 5, 7, 9) at baseline revealed disparate findings compared to analysis of the TME at baseline (Fig. 3). We postulated that these differences are due to resting T cell populations in the periphery versus antigen challenged T cell populations in the TME. We evaluated the functionality of circulating T cells using ex-vivo stimulation with PMA/ionomycin (Fig. 6I). At baseline, the frequency of CD8+ PD1− T cells producing interferon-gamma after stimulation was not significantly different between progressors and non-progressors (median: P 35.3% vs. NP 18.2%, $p = 0.880$; Fig. 6J). After a single cycle of ICI, the frequency of interferon-gamma production post-stimulation was significantly higher in patients with FFP (median: P 16.2% vs. NP 35%, $p = 0.026$; Supplemental Fig. 6J, K). Similarly, at baseline, TNF-α production was not significantly higher in progressors (median: P 61.3% vs. NP 54.1%, $p = 0.530$; Fig. 6L), but after cycle 1 the functional capacity of these

T cells declined in progressors and increased in non-progressors (median: P 49% vs. NP 54.9%, $p = 0.130$; Fig. 6L, M). Likewise, at baseline, interferon production of stimulated CD8+ PD1+ T cells was similar between progressors and those with FFP at baseline (median: P 37.7% vs. NP 47.3%, $p = 0.930$; Fig. 6N), but after one cycle of ICI non-progressors had a significantly higher frequency of T cells producing interferon-gamma upon stimulation (median: P 40.6% vs. NP 53.5%, $p = 0.020$; Fig. 6N, O). Stimulation induced TNF-α production in CD8+ PD1+ T cells at baseline was significantly higher in progressors vs. non-progressors (median: P 82.2% vs. NP 52%, $p = 0.028$; Fig. 6P). After ICI, the frequency of TNF-α + T cells declined significantly in progressors in comparison to baseline values, whereas TNF-α production remained stable in non-progressors, resulting in a loss of the differential seen at baseline (median: P 49.2% vs. NP 51.8%, $p = 0.120$; Fig. 6P, Q). Overall, these results suggest that T cell functionality in response to stimulation increases in non-progressors (but not progressors) after ICI.

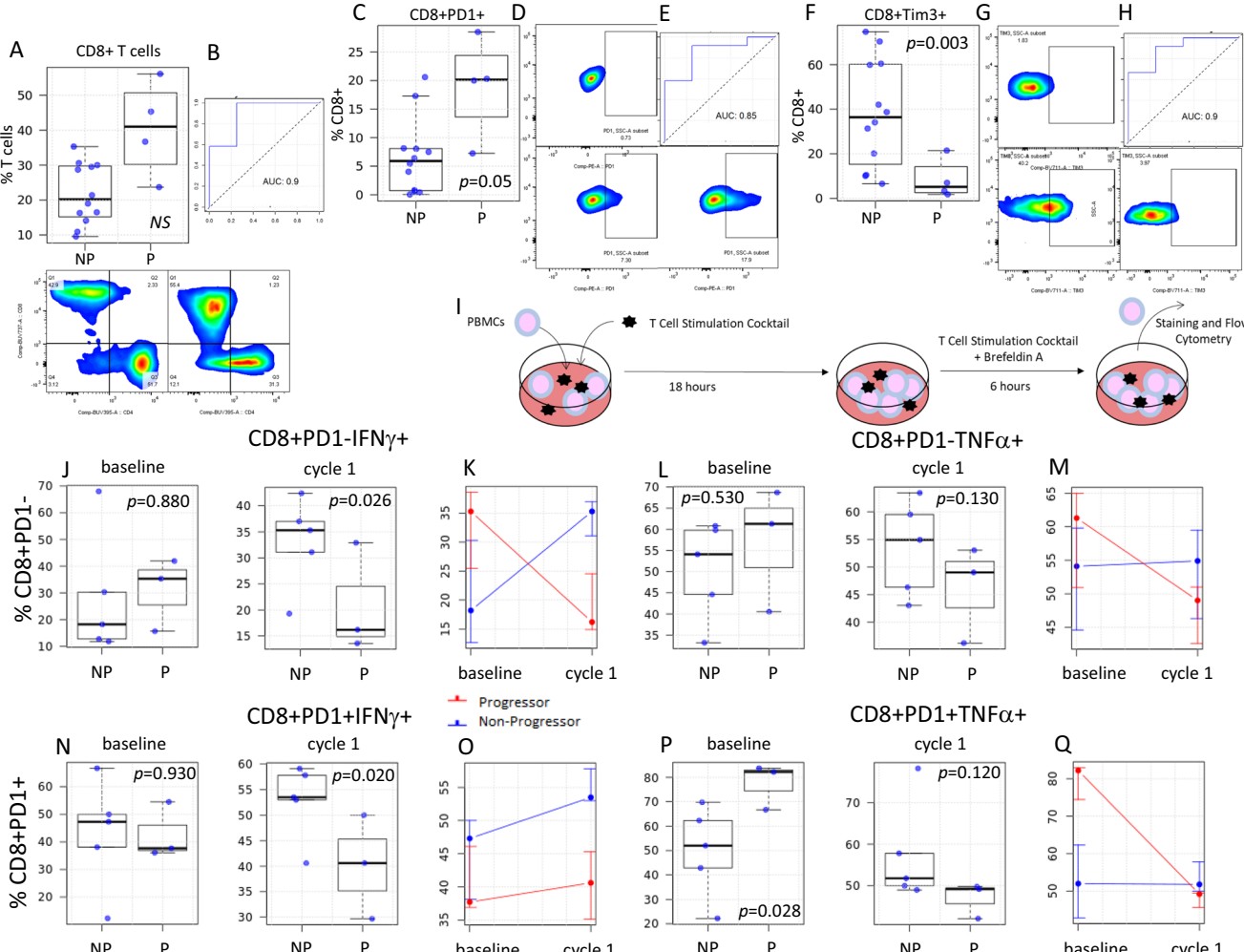

**Fig. 6 | Differences in CD8+ T cell functionality in progressors versus non-progressors. A** Box and whisker plots demonstrating differences in the frequency of circulating CD8+ T cells between progressors (P, $n = 4$ patients) and non-progressors (NP, $n = 12$ patients) at baseline. The panel below depicts representative flow staining for CD4+ T cells and CD8+ T cells on CD3+ gated lymphocytes in NP (left panel) and P (right panel). **B** ROC curve evaluating the ability to classify P vs. NP based on baseline levels of circulating CD8+ T cells. **C** Box and whisker plots demonstrating differences in the frequency of circulating CD8+ PD1+ T cells in P vs. NP at baseline ($n = 16$). **D** Representative flow staining for PD1 on CD8+ gated T cells in FMO negative control (top), NP (bottom left), and P (bottom right). **E** ROC curve evaluating the ability to classify P vs. NP based on baseline levels of circulating CD8+ PD1+ T cells. **F** Box and whisker plots demonstrating differences in the frequency of circulating CD8+Tim3+ T cells in P vs. NP at baseline ($n = 16$). **G** Representative flow staining for Tim3 on CD8+ gated T cells in FMO negative

control (top), NP (bottom left), and P (bottom right). **H** ROC curve evaluating the ability to classify P vs. NP based on baseline levels of circulating CD8+ Tim3+ T cells. **I** Schema of ex-vivo T cell stimulation assay. Box plots and line graphs (left depicting the frequency of interferon-gamma (**J, K, N, O**) and TNF-α (**L, M, P, Q**) after ex-vivo PMA/ionomycin stimulation on PD1- (**J–M**) and PD1+ (**N–Q**) CD8+ T cells collected at baseline or after 1 cycle of ICI ($n = 8$ per timepoint; NP, $n = 5$; P, $n = 3$). For box and whisker plots, each dot represents the value for an individual patient, the line represents the median, the box represents the interquartile range, and the whiskers represent the spread of the data. For line graphs, dots represent the mean, and error bars represent the standard deviation. Patients free from progression are represented in blue, and patients who progressed are represented in red. *P* values are from a two-sided *t*-test, * = $p < 0.05$. Source data are provided as a Source Data file.

## Discussion

Our results demonstrate the tolerability, safety, and feasibility of adding full-dose atezolizumab to SABR in a frail, medically inoperable, early-stage NSCLC patient population. This study was not powered for robust estimates of PFS or OS, and our eligibility criteria were more liberal than most trials in this space, so a direct comparison of treatment outcomes to other trials is inappropriate. However, in our patient population with extensive comorbidities and high-risk factors, the observed median PFS of almost 26 months is encouraging in patients generally not suitable for systemic chemotherapy. Most patients expired from intercurrent illness without evidence of disease progression, limiting the utility of PFS. Thus, FFP was also analyzed post hoc to better evaluate progression. Evaluation of FFP needs to be interpreted with caution as this analysis was post-hoc and did not use a

competing risks approach introducing potential biases into the KM estimates[31]. Median FFP was not reached (Fig. 2A). As there was no control group (SABR alone), we examined an atezolizumab dose–response effect as a potential indicator of activity. Half of the patients were treated at dose level 1 or 2, which is below the approved dose for advanced-stage NSCLC. More than 50% of patients at dose level 1 or 2 progressed, but no patients at full dose (dose level 3) atezolizumab progressed. Patients at dose level 3 had significantly improved FFP (Fig. 2B). Another indicator of activity was a 17% rate of early radiographic response by RECIST after two cycles of atezolizumab prior to SABR. All early responses occurred at dose level 3.

ICIs have become a standard-of-care front-line treatment for metastatic and locally advanced NSCLC following randomized phase III trials demonstrating survival benefits as compared to chemotherapy

alone[1,3,4]. Data in surgically resected NSCLC from the phase III Impower010 and Checkmate 816 trials[5,6] support the integration of ICI therapy into earlier-stage disease. The primary rationale for adding systemic therapy in early-stage, medically inoperable patients is to prevent progression and increase the cure rate. ICI is particularly appealing in this setting, given the limited systemic therapy options in these frail patients and the potential for synergy with standard-of-care SABR.

Preclinical studies demonstrate synergy between radiotherapy and ICI, and some studies find that synergy is suboptimal if ICI is initiated after the completion of radiotherapy[32,33]. Clinical data confirming the utility of combining radiotherapy and ICI have been equivocal, but trials in NSCLC, such as PACIFIC[4], a re-analysis of KEYNOTE-001[24], and a randomized phase II trial by Formenti and colleagues[34] suggest a benefit. Also, a phase II study comparing pembrolizumab alone vs. SABR + pembrolizumab in metastatic NSCLC, while not meeting its endpoint, doubled the ORR from 18% to 36% ($p = 0.07$)[35]. Clinical data also corroborate that the timing of therapy may be key to efficacy. In the PACIFIC trial, patients receiving durvalumab within 14 days of radiotherapy derived greater benefits than those with a longer interval[4]. Another study of 758 patients found that initiation of ICI at least 30 days prior to radiation with continuation during radiation was predictive of increased survival[36]. Our study also suggests the efficacy of SABR and ICI, and the timing of therapy was modeled after the above data.

Our study incorporated robust translational studies in a hypothesis-generating manner to identify potential biomarkers for future clinical trials. A limitation of our correlative analyses was the small patient numbers in this phase I trial, which limits the conclusions that can be drawn. These potential biomarkers and mechanistic findings require validation in larger phase III trials that are underway. PD-L1 expression in the TME is a highly validated predictive biomarker for ICI efficacy in advanced-stage NSCLC. Likewise, the presence of dormant/dysfunctional T cells in the TME can predict ICI response in advanced stage NSCLC[26–28]. We did not observe an association between PD-L1 expression and clinical outcomes (Fig. 1D, Supplemental Fig. 2A). TILs demonstrated importance in our study, but TIL functionality, rather than dysfunction, predicted improved outcomes (Fig. 3).

There is a lack of reliable peripheral blood biomarkers predictive of ICI response. We found differing pre-treatment levels of circulating T cell subsets and transcriptomic signatures of T cell activation between patients who progress versus those free from progression (Figs. 5 and 6). Using ROC analysis (Fig. 6, Supplemental Fig. 9), we demonstrated that many of these markers are highly predictive for distinguishing progressors vs. non-progressors. Surprisingly, at baseline, circulating T cell frequencies, PD1 expression, and transcriptomic signatures of activation were higher in progressing patients. This is in contrast to the TME, where T cell frequencies and function were higher in patients free from progression and at odds with published data in advanced malignancies where high circulating T cell levels and PD1+ expression correlate with ICI response. Interestingly, while the frequency of circulating T cell subsets and PD1 expression on these were higher in progressors, the frequency of Tim3+ T cell subsets was higher in patients without progression. One possible explanation for these observations, and the differences between the TME and peripheral T cells, is that T cells in the TME are largely tumor antigen specific[37] whereas those in the periphery are largely resting and not tumor antigen-specific. Tim3 expression is thought to identify T cells that are antigen-specific in chronic viral infections and cancer[38]. Thus, Tim3+ T cells in the periphery may better mirror the phenotype of T cells in the TME, but additional studies are needed to test this hypothesis.

To further explore these findings, we tested T-cell functionality after ex-vivo stimulation. After one cycle of ICI, stimulated T cell functionality increased in non-progressors and decreased in progressors compared to pre-treatment values (Fig. 6). This supports the idea that the functionality capacity of T cells and the ability to be re-invigorated after ICI are important for favorable clinical outcomes in this population. Taken together, our correlative findings suggest that while T cell dysfunction and a suppressive TME are important ICI biomarkers in advanced NSCLC, in early-stage NSCLC, T cell functionality is strongly linked to treatment outcomes. The differences in our findings, as compared with prior reports, could be attributed to a lack of statistical power or a genuine difference in biology. The highly significant association between clinical outcomes and correlative parameters, despite limited patient numbers, argues for the latter. Genuine biological differences in ICI mechanisms of action are plausible due to a different immune landscape in early-stage vs. advanced NSCLC and/or in this relatively immunocompromised population. These findings are hypothesis-generating and will be tested in a larger randomized phase III trial. Additionally, many other novel biomarkers currently under study, such as a recent preclinical report of putative secreted biomarkers for combined radiotherapy + immunotherapy in NSCLC, will also need to be considered in future trials[39].

This study regimen has been adapted into the phase III randomized SWOG/NRG Oncology S1914 trial, currently active to accrual (NCT04214262). S1914 randomizes patients between standard-of-care SABR and SABR with a total of 8 cycles of neoadjuvant, concurrent, and adjuvant atezolizumab. A total of 480 patients are planned to be randomized, with a primary endpoint of OS. This study will also test the correlative biomarkers reported here, and the presence of a control arm will allow the determination of whether these biomarkers are prognostic for early-stage NSCLC in general or predictive of response to atezolizumab. Two other ongoing randomized phase 3 trials are also testing the integration of ICI and SABR for medically inoperable, early-stage NSCLC, PACIFIC-4 (NCT03833154), and KEYNOTE-867 (NCT03924869).

## Methods
### Patient eligibility
The study was authorized and approved by the IRB of, and patients were recruited from three centers: The University of California Davis Comprehensive Cancer Center, Cedars Sinai Medical Center, and VA David Grant Medical Center. The study design and conduct complied with all relevant regulations regarding the use of human study participants and was conducted in accordance with the criteria set by the Declaration of Helsinki. All patients provided written informed consent prior to any study procedures. The trial was registered on clinicaltrials.gov (NCT02599454). The first patient was enrolled on 5/3/2016, and the last patient was enrolled on 8/9/2019. Eligible patients were ≥18 years of age with histologically confirmed T1-3 NSCLC ≤ 7 cm diameter. Although patients with EGFR or ALK mutations would not be expected to respond to ICI based on data from advanced NSCLC, these patients were not excluded since there is limited tissue available for NGS testing in this population. Patients with chest wall invasion (T3) and 2 nodules within the same lobe of the lung were eligible. Patients were required to have one or more features predictive of increased recurrence risk: diameter ≥1 cm for the phase I component and ≥2 cm for the expansion cohort; SUV ≥ 6.2 on FDG PET; or moderately/poorly differentiated histology. Patients were required to be either medically inoperable as determined by multidisciplinary evaluation or to have refused surgery, had a forced expiratory volume over 1 s (FEV1) ≥ 700 cc, and a diffusing capacity for carbon monoxide (DLCO) ≥ 5.5 m/min/mm Hg on pulmonary function testing (PFT), and had a Zubrod performance status (PS) ≤ 2. Exclusion criteria included New York Heart Association class II or greater cardiovascular disease, severe infections within 4 weeks of enrollment, history of autoimmune disorders, idiopathic pulmonary fibrosis, and active human immunodeficiency virus, hepatitis B, or hepatitis C. Prior malignancies did not disqualify as long as they were not active at the time of enrollment. Required

staging workup included PFTs within 3 months of registration and computed tomography (CT) of the chest within 28 days of registration. PET/CT staging was not mandated but was encouraged.

## Study design

The trial was designed as a proof-of-concept phase I study with a standard 3 + 3 dose escalation design followed by a patient expansion cohort (Fig. 1A). Atezolizumab was delivered intravenously (IV) in 21-day cycles. Three atezolizumab dose levels were evaluated: 3 mg/kg IV, 10 mg/kg IV, and 1200 mg IV flat dosing (Fig. 1A). Patients received a planned 6 cycles of atezolizumab with SABR initiated with cycle 3, 24–48 h following the atezolizumab infusion. SABR was delivered to 50 Gy over 4–5 fractions. DLT was defined as ≥grade 3 immune-related adverse event or other ≥grade 3 treatment-related adverse events that did not resolve to ≤grade 2 within 14 days of onset, or grade 2 diarrhea, aspartate aminotransferase (AST)/alanine transaminase (ALT) > 3× the upper limit of normal with bilirubin > 2× upper limit of normal, or pneumonitis that required holding treatment >14 days. The DLT period was 9 weeks. Patients were assessed with labs, including a completed blood count, metabolic panel, c-reactive protein, and thyroid function tests prior to each cycle. Tumor assessment was performed with either PET/CT or CT at the discretion of the treating physician every 2 cycles during treatment. After completion of treatment, tumor assessment was performed every 3 months, year 1–2, and every 6 months, year 3–5. The full trial protocol is included in the Supplementary information file.

## Stereotactic ablative radiotherapy

All patients underwent CT simulation with slice thickness ≤3.0 cm with reliable immobilization. Four-dimensional (4D) CT simulation was strongly encouraged but not required. For patients simulated with 4DCT, an internal target volume (ITV) was created using the 10 respiratory phases, and a 5 mm planning target volume (PTV) margin was added in all directions. For patients simulated without 4DCT, a 5 mm PTV margin was added in the axial plane, and a 1.0–1.5 cm margin craniocaudally was added depending on tumor motion as assessed by fluoroscopy. A motion management strategy (abdominal compression, respiratory gating, tumor tracking, or breath-hold) was required if respiratory motion, as assessed by fluoroscopy, exceeded 1 cm. SABR was delivered to 50 Gy over 4 fractions for peripherally located tumors (>2 cm from the proximal bronchial tree and not touching mediastinal pleural) and to 50 Gy over 5 fractions for centrally located tumors. Fractions were delivered 40–96 h apart. The prescription isodose surface was chosen such that 95% of the PTV was covered by the prescription isodose line, and 99% of the PTV received a minimum of 90% of the prescription dose. Dose constraints included combined lung-GTV volume receiving 20 Gy (V20) < 10%, with less than 1500 cc combined lung receiving 12.5 Gy and less than 1000 cc combined lung receiving 13.5 Gy. Full normal tissue dose constraints are provided in Supplemental Table 1.

## Outcomes

The primary objective was to determine the maximum tolerated dose (MTD) of atezolizumab when delivered with SABR in early-stage, medically inoperable NSCLC. MTD was defined as the highest dose at which no more than one of six patients developed a DLT or dose Level 3 if the MTD was not reached. Secondary objectives included the safety profile of the experimental regimen using common toxicity criteria for adverse events (CTCAE) version 4 and preliminary efficacy data by objective response rate (ORR) and PFS by response evaluation in solid tumors (RECIST) version 1.1[40]. This patient population is at very high risk for developing additional primary aero-digestive tract malignancies due to "field cancerization" effects. In order to distinguish the development of a new primary NSCLC from local or systemic disease progression, the development of disease in the contralateral lung without evidence of ipsilateral or systemic recurrence was reviewed and characterized by a multidisciplinary tumor board. In addition to PFS, OS, and FFP were analyzed. FFP (time from enrollment to documentation of disease progression) was a post-hoc analysis undertaken because of the significant rate of death from an intercurrent illness unrelated to cancer or study treatment. Survival and progression were measured from the day of trial enrollment. ORR was assessed following cycle 2, prior to initiation of SABR, as the lung changes post-SABR make response assessment inaccurate.

## Correlative science tissue collection

Baseline tumor samples were required for study participation and were taken from tissue blocks or fresh tumor biopsies. Samples were formalin-fixed and paraffin-embedded (FFPE), and fresh sections of FFPE were used for immunohistochemistry (IHC) and multi-plex immune fluorescence (IF). Blood samples were collected at baseline, after every cycle, and end of treatment. Peripheral blood mononuclear cells (PBMCs) and plasma were isolated from the whole blood. PBMCs were cryopreserved for batched analysis. Stool samples were collected at baseline, end of cycle 2, and end of treatment. Determining response by RECIST criteria after radiotherapy ablation in NSCLC is difficult to interpret and is generally not performed as radiotherapy scar tissue cannot be distinguished from tumors. This makes applying RECIST criteria and determining response rates after lung SBRT problematic in this population which only has a single lesion to evaluate for a response. As an indicator of ICI efficacy, early response to ICI alone (before SBRT) is reported. Post-ablation imaging was not used to determine or confirm responses.

## Tissue analysis

Tumor samples were stained for PD-L1 by immuno-histochemistry on the Ventana Discovery Ultra autostainer (Roche) using the clone E1L3N (cell signaling) at a dilution of 1:100 as previously described[41]. Briefly, "4 μm FFPE sections were mounted on Superfrost Plus microscope slides (Thermo Fisher Scientific) and dried overnight. Sections were deparaffinized, followed by antigen retrieval in CC1 buffer (pH 9, 95 °C; Roche), endogenous peroxidase blocking, and then incubation with the primary antibodies. Chromogenic detection was performed with Chromomap DAB (Roche), followed by counterstaining with hematoxylin. Sections stained with and without primary antibody were used as positive and negative controls." Membrane staining was verified and quantified by a board-certified pathologist blinded to outcomes using TPS as previously described[42]. Additional sections, when available, were also stained and analyzed by multi-plex IF for CD3, Ki-67, and granzyme b (GZB) as previously described[26]. Briefly, "FFPE histology sections were deparaffinized and subjected to antigen retrieval using EDTA buffer (Sigma-Aldrich) pH = 8.0 and boiled for 20 min at 97 °C in a pressure-boiling container (PT module, Lab Vision). Slides were then incubated with dual endogenous peroxidase block (DAKO #S2003) for 10 min at room temperature and subsequently with a blocking solution containing 0.3% bovine serum albumin in 0.05% Tween solution for 30 min. Slides were stained with 4′,6-Diamidino-2-Phenylindole (DAPI) for visualization of all cells, CK to detect tumor epithelial cells, CD3 for T-lymphocytes, GZB for T-cell cytolytic potential and Ki-67 as a cell proliferation marker. Primary antibodies included CK clone AE1/AE3 (catalog # M3515) from DAKO used with a concentration of 0.12 mg/ml, CD3 clone SP7 (catalog # NB600-1441) from Novus biologicals dilution 1:100 (culture supernatant), GZB clone 4E6 (catalog # ab139354) from Abcam with a concentration of 5 μg/ml and Ki-67 clone MIB1 (catalog # M724029-2) from DAKO with a concentration of 0.46 μg/ml. Secondary antibodies and fluorescent reagents used were goat anti-rabbit Alexa546 (Invitrogen; 1:100 dilution), anti-rabbit Envision (K4009, DAKO, 1:100 stock dilution) with biotinylated tyramide/Streptavidine-Alexa750 conjugate (Perkin-Elmer); anti-mouse IgG1 antibody (eBioscience) with fluorescein-tyramide (Perkin-Elmer), anti-mouse

IgG2a antibody (Abcam) with Cy5-tyramide (Perkin-Elmer). Residual horseradish peroxidase activity between incubations with secondary antibodies was eliminated by exposing the slides twice for 7 min to a solution containing benzoic hydrazide (0.136 mg) and hydrogen peroxide (50 μl)." Fluorescence was quantified using the AQUA method, and the QIF score was calculated by dividing the target pixel intensities by the area of positivity in the sample. Slides were examined by a pathologist to exclude defective areas and staining artifacts.

### Flow cytometry and ex-vivo stimulation assays

In this report, we focus on samples obtained pre-treatment and after cycles 1–3 to discover potential early biomarkers of clinical outcomes. Analysis focusing on additional cycles and the effects of SABR on the immune response will be reported elsewhere. PBMCs were immuno-phenotyped using flow cytometry. Briefly, PBMCs were thawed and incubated with Fc-block (BD Bioscience, Franklin Lakes, NJ) on ice for 15 min. Then, cells were stained with specific antibody cocktails (supplemental Table 2) for 1 h on ice and then stained with Aqua-LIVE/DEAD (Invitrogen, Carlsbad, CA) for 30 min at room temperature. Cells were washed after each step using PBS containing 0.5% BSA and before being processed on a BD Fortessa flow cytometer (BD Bioscience, Franklin Lakes, NJ). Data were analyzed using FlowJo software version 10.6.2 (Tree Star Inc. Ashland, OR). Selected samples were stimulated ex-vivo (see supplemental figure 8A) using eBioscience stimulation cocktail (Invitrogen, Carlsbad, CA). Cytokine secretion was prevented using eBioscience Brefeldin A (Invitrogen, Carlsbad, CA). Cells were then stained with a specific antibody cocktail (Supplemental Table 2). Intracellular cytokine staining used the eBioscience FOXP3/Transcription Factor Staining Buffer Set (Invitrogen, Carlsbad, CA) according to the manufacturer's protocol.

### Transcriptomic analysis

RNA was extracted from cryopreserved PBMCs and sequenced for RNA expression and TCR analysis[43]. Total RNA was extracted using a Quick-RNA MagBead kit (Zymo Research) from cryopreserved PBMC. RNA concentrations were then quantified using a Qubit Fluorometer (Invitrogen), and RNA integrity was assessed using the Agilent TapeStation (Agilent).

Indexed libraries were constructed using the SMARTer® Stranded Total RNA-Seq Kit v3 (Takara Bio) following the manufacturer's instructions. The quantity and quality of the libraries were assessed by Qubit and Agilent 2100 Bioanalyzer, respectively. Libraries' molar concentrations were validated by qPCR for library pooling. Sequencing was performed on the Illumina HiSeq 4000 platform using PE150 chemistry (Illumina).

The raw data were aligned to the hg38 reference genome, and the number of reads per gene was counted using STAR v.2.5.1[44]. The DESeq2 R package[45] was applied for differential expression analysis and gene expression normalization. All p values were corrected using the false discovery rate (FDR) method. Genes with corrected p-values less than 0.05-, and two or more-fold change differences are considered differentially expressed. Gene expression data were log-transformed for principal components and clustering analysis. The clustered heatmap was created with the R package "pheatmap"[46].

### Statistics and reproducibility

The clinical trial was a non-randomized phase I trial. For the dose escalation component, a traditional 3 + 3 design was used. By study design, 3–6 patients were expected to enroll. An additional 15-patient expansion cohort was planned. With 6 patients treated at the MTD in the dose escalation phase and an additional 15-patient expansion cohort, a minimum of 21 patients were expected to be available for assessing safety and estimating efficacy at the MTD. The trial closed early following the accrual of 5 patients to the expansion cohort due to activation of the phase 3 SWOG/NRG S1914, which enrolled this population. A total of 20 patients were enrolled. Descriptive statistics (including frequency, proportion, median, and range) were used to summarize patient demographics, tumor characteristics, toxicity, and response status. A waterfall plot was generated to describe the relative changes in tumor size using R version 4.0.4. OS, PFS, and FFP were analyzed by the Kaplan–Meier method[47] and compared between subgroups by log-rank tests using GraphPad Prism version 8.3.

The sample size for correlative studies was based on the presence of sufficient material to perform the studies. No statistical method was used to predetermine the sample size for correlative studies. No data were excluded from the analyses. This trial did not include randomization, and investigators were not blinded during experiments and outcome assessment. Statistical analysis of IHC/IC data was performed using descriptive statistics. Comparisons across two groups (e.g., responders vs. non-responders or non-progressor vs. progressor) were performed using a two-sided $t$-test. Data were analyzed using Graph-Pad Prism version 8.3. For the evaluation of PBMCs and transcriptomic, a two-sided $t$-test was used to compare two groups, and ANOVA was used to evaluate multiple groups or time points. Flow cytometry data were analyzed using Flowjo software (v. 6.10.2). Percent cells were examined for normality and log transformed when necessary. Statistical analyses were performed using the R statistical package (v.4.1.1)[48]. A two-sided $t$-test was used to compare two groups of patients, and $p$-value less than 0.05 were considered statistically significant. Since these analyses were hypothesis-generating in nature to discover potential biomarkers that will be further analyzed in an ongoing phase III trial, we did not control for multiple comparisons (with the exception of transcriptomic analyses). Potential biomarkers were analyzed using logistic regression implemented in the R package logistf[49]. The logistic regression ROC analysis was performed with the "ROCR" package[50].

### Reporting summary

Further information on research design is available in the Nature Portfolio Reporting Summary linked to this article.

## Data availability

The RNA sequencing data generated in this study have been deposited in the sequence read archive (SRA) database under accession code PRJNA999721. Individual de-identified participant data on demographics, toxicity, and clinical outcomes, will be maintained by the UC Davis Cancer Center Office of Clinical Research and will be shared for academic purposes on request (Dr. Megan E. Daly, medaly@ucdavis.edu) for at least two years from the date of publication, with the completion of a data access agreement. Individual de-identified data for correlative studies and transcriptomics is included in the manuscript or publicly available as listed above. Multiplex IHC imaging data is maintained by Dr. Kurt Schalper (kurt.schalper@yale.edu) and will be shared for academic purposes on request for at least 2 years from the date of publication. The study protocol is available as Supplementary Note in the Supplementary Information file. Source data are provided in this paper. The remaining data are available within the Article, Supplementary Information, or Source Data file. Source data are provided in this paper.

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

## Acknowledgements

Source of funding: This work was supported by the Office of the Assistant Secretary of Defense for Health Affairs through the Lung Cancer Research Program under Award No.W81XWH-15-2-0063. Atezolizumab was provided by Genentech. Institutional Support was provided by the UC Davis Comprehensive Cancer Center Support Grant NCI P30CA093373. AMM and LVV received support through NIH/NCI R01 CA240751. H.M.M. received support through NIH/NCI 4R00CA256526. Role of the funding source: These funding sources had no role in study design; in the collection, analysis, and interpretation of data; in the writing of the report; and in the decision to submit the paper for publication.

## Author contributions

A.M.M.—conceptualization, data curation, formal analysis, funding acquisition, investigation, methodology, project administration, resources, software, supervision, validation, visualization, writing—original draft, and writing—review & editing. M.E.D.—conceptualization, data curation, formal analysis, funding acquisition, investigation, methodology, project administration, resources, software, supervision, validation, visualization, writing—original draft, and writing—review & editing. G.L.—formal analysis, software, resources, investigation, methodology, writing—original draft, and writing—review & editing. E.M.—supervision, formal analysis, software, resources, investigation, methodology, and writing—review & editing. A.A.M.—formal analysis, software, investigation, methodology, A.I.M.—formal analysis, software, investigation, methodology, A.B.—formal analysis, software, resources, investigation, methodology, A.M.—investigation, supervision, project administration. S.L.S.—formal analysis, resources. L.B.—data curation, formal analysis, investigation, supervision, project administration, writing—review & editing. S.C.—data curation, formal analysis, investigation, supervision, project administration, writing—review & editing. D.E.—investigation, supervision, project administration. T.L.—investigation, supervision, project administration. L.V.V.—formal analysis, writing—original draft, and writing—review & editing. H.M.M.—formal analysis, writing—original draft, and writing—review & editing. F.L.—data curation, formal analysis, project administration. L.G.—data curation, formal analysis, project administration. L.A.M.—data curation, formal analysis, project administration. R.J.C.—formal analysis, and writing—review & editing. J.W.R.—investigation, supervision, project administration, writing—review & editing. K.A.S.—supervision, formal analysis, software, resources, investigation, methodology, and writing—review & editing. W.J.M.—supervision, resources, methodology, and writing—review & editing. K.K.—conceptualization, data curation, formal analysis, funding acquisition, investigation, methodology, project administration, resources, software, supervision, validation, visualization, writing—original draft, and writing—review & editing.

## Competing interests

A.M.M. – Grants/Contracts: Incyte, Merck, Genentech, BMS, Transgene, EMD Serono, Trisalus. Consulting Fees: Atheneum, First Thought, Opinion Site, Alcimed. Honoraria: ANCO, ACVR. Advisory Board and Stock Options: Multiplex Thera. M.E.D.—Grants/Contracts: Merck, Genentech, EMD Serono. Consulting Fees: Astra Zeneca. Honoraria: Curio, Dava. G.L.—none. E.M.—none. A.A.M.—none. A.I.M.—none. A.B.—none. A.M.—none. S.L.S.— none. L.B.—none. S.C.—none. D.E.—none. T.L.—none. L.V.V.—none. H.M.M.—Consulting fees: RefleXion. F.L.—none. L.G.—none. L.A.M.—none. R.J.C.—Advisory Board: NAKI Therapeutics. J.W.R.—Grants/Contracts: AstraZeneca, Boehringer Ingelheim, Merck, Novartis, Revolution Medicines, Spectrum. K.A.S.—Grants/Contracts: Navigate Biopharma, Tesaro/GSK, Moderna Inc., Takeda, Surface Oncology, Pierre-Fabre Research Institute, Merck, Bristol-Myers Squibb, AstraZeneca, Ribon Therapeutics, Eli Lilly, Boehringer-Ingelheim and Akoya Biosciences. Consulting Fees: Clinica Alemana Santiago, Shattuck Labs, AstraZeneca, EMD Serono, Takeda, Torque/Repertoire Therapeutics, Agenus, Genmab, OnCusp, Parthenon Therapeutics, Bristol-Myers Squibb, Roche, CDR life, Sensei Therapeutics, Molecular Templates and Merck. Honoraria: Takeda, Fluidigm, Merck, Brstil Myers Squibb, PeerView, Forefront collaborative. W.J.M.—Grants/Contracts: Merck. K.K.—Grants/Contracts: Genentech, BMS, Transgene. Advisory Board: Genentech. Other: IASLC. A.M.M., M.E.D., and K.K. received support from Genentech in the form of providing Atezolizumab for this clinical trial. Genentech stand to potentially benefit financially from this report. Genentech played no role in the conceptualization, design, data collection, analysis, decision to publish, or preparation of the paper.
