## [Peer Review File · Nature Communications]

Atezolizumab plus Stereotactic Ablative Radiotherapy for Medically Inoperable Patients with Early-Stage Non-Small Cell Lung Cancer: A Multi-Institutional Phase I trialEditorial Note: This manuscript has been previously reviewed at another journal that is not operating a transparent peer review scheme. This document only contains reviewer comments and rebuttal letters for versions considered at *Nature Communications*. Mentions of the other journal have been redacted.

REVIEWER COMMENTS

Reviewer #1 (Remarks to the Author):

This is a re-review of the manuscript by Monjazez et al. on a phase I clinical trial examining 20 patients with treated with atezolizumab (atezo) and SBRT for non-resectable (or patient refusal for surgery) for T1-T3NxM0 NSCLC accrued over 3 years. The manuscript is now being considered for publication in Nature Communications and the authors have responded to prior Reviewer comments to the original submission to [redacted].

The authors were as responsive to the Reviewer's comments as they could be given the significant limitations of a small phase I trial, limited correlative data points, and the inability to have any post-treatment biopsy or use RECIST assessment after SBRT. I appreciate that the authors continue to believe that emphasizing outcomes data are important, but that is not the focus of a phase I trial. In addition, linking correlatives to very limited outcomes may be hypothesis-generating, but showing KM curves and statistics seems overly ambitious in this effort.

With respect to the authors assertion that radiographic response to single agent IO therapy is an acceptable measure of response I do not agree. The majority (all?) of neoadjuvant IO trials to date have found that radiographic response is not a good indicator of pathologic response – I do not think it would be any different just because the tumor is in a frail, inoperable patient. In fact, one could argue that these patients are not that “frail” given that 85% have a Zubrod performance status of 0-1 with 40% being Zubrod 0. Median FEV1 was 61% and DLCO was not even reported. How many patients refused surgery vs. were considered inoperable based on clinical assessment?

I appreciate the clarification on the transcriptomic analysis and use of the BIOCARTA T-cell receptor signaling pathway.

Overall, the manuscript has been modified minimally since the original submission. As noted previously and again in this review the results show that patients can tolerate this treatment. Unfortunately, the efforts to link certain correlative studies to outcomes continues to be an overreach in this Reviewer's opinion (this was also mentioned in Reviewers 2 comments).

Reviewer #3 (Remarks to the Author):

Dear Authors,

I appreciated the responses to each comment. All comments have been adequately addressed, but I have additional comments on items 1 and 2 below.

1. The definition of FFP is not clearly defined in the manuscript. For example, it is unclear if patients who died without progression were censored. If so, it is well-known that this approach leads to incorrect and biased results (see details in Kim, 2007).

Response: We agree with and appreciate the reviewer pointing out this bias and we have now included a discussion of this bias and the reference in the manuscript (line 462-463). We have also clearly defined FFP in the methods (line 204). Despite the potential bias in FFP we feel it is clinically relevant in this particular population since the majority of patients who expired did so without evidence of disease progression or treatment related toxicity but rather of intercurrent illness. The addition of FFP alongside of OS and PFS allows better interpretation of the data since each endpoint has its own limitations / biases. In populations where death from intercurrent illness rates can be very high cause specific survival and/or freedom from progression are often reported in addition to OS and PFS (for example see pmid 35569466). We agree that FFP has biases, cannot replace OS and PFS, and we report all three endpoints.

The primary rationale of adding ICI to SBRT is to prevent progression and thus the ad-hoc analysis of FFP was undertaken. The correlative studies compare progressors vs those free

from progression as a binary endpoint.

Comments: I agree that FFP is an important endpoint but respectfully disagree with the response because the unbiased progression rate can be estimated using the competing risk approach. In addition, the discussion on the bias and the definition of FFP is not provided in lines 462-463 and 204.

2. In lines 500-501, the authors claim that using ROC analysis, the biomarkers shown in Figure 6 and Supplemental Figure 9 are highly predictive of FFP. However, it is unclear if the binary outcomes (progressors vs. non-progressors) were used to estimate the area under the ROC curve. If so, this approach excluded the censored subjects and time information. Thus, the authors need to rephrase the sentence as appropriate.

If the FFP was treated as the time-to-event variable, please provide the method that was used for computing the ROC value.

Response: We apologize for this error. As the reviewer points out this is an analysis of binary outcomes and the sentence has been updated to clarify (line 503).

Comments: Please clarify it in the manuscript since line 503 doesn't have it.

Reviewer #4 (Remarks to the Author): with expertise in lung cancer, therapy, clinical
Thank you for the opportunity to review this paper – I have been invited as an additional reviewer and note the changes made to the manuscript made to address the R1 review.

In essence this is a comprehensive translational analysis into a phase I trial. Despite the limited numbers as expected in a phase I dataset, the depth of the translational analysis is commendable. I do not agree with several of the comments from reviewer #1 questioning the novelty of the analysis or the lack of tissue or control arm. Combination SABR and ICI studies in this early-stage medically inoperable cohort is very limited, and tissue acquisition in particular is extremely difficult. This is not a stage IV cohort or an operable neoadjuvant cohort, where tissue is expected to be more abundant. Overall I think this study is

commendable, despite its limitations in size and hypothesis-generating nature.

Further comments

The authors make an effort to describe the definition of new primaries but don't clarify why – is this because new primary events were discounted from PFS? One of the progressions were in the contralateral lung - was this defined as a new primary or part of PFS?

Need to clarify in discussion and results that 'ORR' is 'ORR to atezolizumab' to ensure no confusion of radiation effect for the reader. This would address one of the other reviewer queries.

The HR for the differences between DL 3 and 1 are very wide (hazard ratio: 0.156 (0.027-0.911)); although statistically significant, it should be acknowledged in the results line 337 that this is exploratory and hypothesis generating.

The translational analyses are comprehensive. Several potential issues in interpretation have been highlighted by the reviewers, however with a phase I trial these findings are intended to be exploratory and thus the responses to the reviewers appear justified.

We thank the reviewers for their time in reviewing our submission. We thank the reviewers for their insightful comments and feel that the manuscript has been strengthened by incorporating their suggestions. We have responded to each reviewer individually below their reviews (response in bold). Also, although throughout the manuscript the conclusions on efficacy and correlative studies are listed as "preliminary" and "hypothesis generating", we have attempted to further tone down the conclusions in response to the reviewers. We have made the general changes outlined below:

Line 332: The freedom from progression analysis has been removed from the outcomes paragraph and placed under a new subheading entitled Post-Hoc Analysis of FFP.

Line 50. The conclusion of the abstract has been changed to: "These results require validation and are being tested in a phase III randomized trial."

Line 101. The last line of the introduction we have added: "Conclusions regarding efficacy and correlative outcomes are limited by small patient numbers and the early-stage nature of this trial and should be viewed as hypothesis generating."

Line 330. We have added the statement: "Efficacy and response data in this small phase 1 trial should be viewed as hypothesis generating."

Line 340: We have added the statement: "This analysis is hypothesis generating and the efficacy of this approach is being tested in a randomized phase III trial."

Line 438: We have removed the statement: "and signals a potential benefit of this approach in preventing progression" from the discussion.

Line 474: We have added the statement: "A limitation of our correlative analyses was the small patient numbers in this phase I trial, which limits the conclusions that can be drawn. These potential biomarkers and mechanistic findings require validation in larger phase III trials which are underway."

Reviewer #1 (Remarks to the Author):

This is a re-review of the manuscript by Monjazez et al. on a phase I clinical trial examining 20 patients with treated with atezolizumab (atezo) and SBRT for non-resectable (or patient refusal for surgery) for T1-T3NxM0 NSCLC accrued over 3 years. The manuscript is now being considered for publication in Nature Communications and the authors have responded to prior Reviewer comments to the original submission to [redacted].

The authors were as responsive to the Reviewer's comments as they could be given the significant limitations of a small phase I trial, limited correlative data points, and the inability to have any post-treatment biopsy or use RECIST assessment after SBRT. I

appreciate that the authors continue to believe that emphasizing outcomes data are important, but that is not the focus of a phase I trial. In addition, linking correlatives to very limited outcomes may be hypothesis-generating, but showing KM curves and statistics seems overly ambitious in this effort.

With respect to the authors assertion that radiographic response to single agent IO therapy is an acceptable measure of response I do not agree. The majority (all?) of neoadjuvant IO trials to date have found that radiographic response is not a good indicator of pathologic response – I do not think it would be any different just because the tumor is in a frail, inoperable patient. In fact, one could argue that these patients are not that “frail” given that 85% have a Zubrod performance status of 0-1 with 40% being Zubrod 0. Median FEV1 was 61% and DLCO was not even reported. How many patients refused surgery vs. were considered inoperable based on clinical assessment?

I appreciate the clarification on the transcriptomic analysis and use of the BIOCARTA T-cell receptor signaling pathway.

Overall, the manuscript has been modified minimally since the original submission. As noted previously and again in this review the results show that patients can tolerate this treatment. Unfortunately, the efforts to link certain correlative studies to outcomes continues to be an overreach in this Reviewer’s opinion (this was also mentioned in Reviewers 2 comments).

RESPONSE: We again thank the reviewer for their time and consideration in reviewing this manuscript. We believe that the hypothesis generating efficacy and correlative findings will be of interest in the field so that they can be tested and validated in larger studies. We do however appreciate the reviewers concern that the conclusions drawn could be overstated and that the primary endpoint of a phase I trial is MTD and toxicity. We have therefor, as outlined above, tried to tone down the conclusions and repeatedly remind the reader that these other endpoints are hypothesis generating, may be biased by limited numbers, and require validation.

Reviewer #3 (Remarks to the Author):

Dear Authors,

I appreciated the responses to each comment. All comments have been adequately addressed, but I have additional comments on items 1 and 2 below.

1. The definition of FFP is not clearly defined in the manuscript. For example, it is unclear if patients who died without progression were censored. If so, it is well-known that this approach leads to incorrect and biased results (see details in Kim, 2007).

Response: We agree with and appreciate the reviewer pointing out this bias and we have

now included a discussion of this bias and the reference in the manuscript (line 462-463). We have also clearly defined FFP in the methods (line 204). Despite the potential bias in FFP we feel it is clinically relevant in this particular population since the majority of patients who expired did so without evidence of disease progression or treatment related toxicity but rather of intercurrent illness. The addition of FFP alongside of OS and PFS allows better interpretation of the data since each endpoint has its own limitations / biases. In populations where death from intercurrent illness rates can be very high cause specific survival and/or freedom from progression are often reported in addition to OS and PFS (for example see PMID 35569466). We agree that FFP has biases, cannot replace OS and PFS, and we report all three endpoints.

The primary rationale of adding ICI to SBRT is to prevent progression and thus the ad-hoc analysis of FFP was undertaken. The correlative studies compare progressors vs those free from progression as a binary endpoint.

Comments: I agree that FFP is an important endpoint but respectfully disagree with the response because the unbiased progression rate can be estimated using the competing risk approach. In addition, the discussion on the bias and the definition of FFP is not provided in lines 462-463 and 204.

2. In lines 500-501, the authors claim that using ROC analysis, the biomarkers shown in Figure 6 and Supplemental Figure 9 are highly predictive of FFP. However, it is unclear if the binary outcomes (progressors vs. non-progressors) were used to estimate the area under the ROC curve. If so, this approach excluded the censored subjects and time information. Thus, the authors need to rephrase the sentence as appropriate. If the FFP was treated as the time-to-event variable, please provide the method that was used for computing the ROC value.

Response: We apologize for this error. As the reviewer points out this is an analysis of binary outcomes and the sentence has been updated to clarify (line 503).

Comments: Please clarify it in the manuscript since line 503 doesn't have it.

RESPONSE: We thank the reviewer for their time and apologize for the confusion regarding the changes that were made. The line numbers referred to the tracked changes version of the manuscript. FFP is defined on line 173 in the materials and methods. The discussion on the bias introduced by FFP and the reference suggested by the reviewer can be found on line 441 in the discussion which has been further expanded upon in this version of the manuscript.

Reviewer #4 (Remarks to the Author): with expertise in lung cancer, therapy, clinical

Thank you for the opportunity to review this paper – I have been invited as an additional reviewer and note the changes made to the manuscript made to address the R1 review.

In essence this is a comprehensive translational analysis into a phase I trial. Despite the limited numbers as expected in a phase I dataset, the depth of the translational analysis is commendable. I do not agree with several of the comments from reviewer #1 questioning the novelty of the analysis or the lack of tissue or control arm. Combination SABR and ICI studies in this early-stage medically inoperable cohort is very limited, and tissue acquisition in particular is extremely difficult. This is not a stage IV cohort or an operable neoadjuvant cohort, where tissue is expected to be more abundant. Overall I think this study is commendable, despite its limitations in size and hypothesis-generating nature.

Further comments

The authors make an effort to describe the definition of new primaries but don't clarify why – is this because new primary events were discounted from PFS? One of the progressions were in the contralateral lung - was this defined as a new primary or part of PFS?

Need to clarify in discussion and results that 'ORR' is 'ORR to atezolizumab' to ensure no confusion of radiation effect for the reader. This would address one of the other reviewer queries.

The HR for the differences between DL 3 and 1 are very wide (hazard ratio: 0.156 (0.027-0.911)); although statistically significant, it should be acknowledged in the results line 337 that this is exploratory and hypothesis generating.

The translational analyses are comprehensive. Several potential issues in interpretation have been highlighted by the reviewers, however with a phase I trial these findings are intended to be exploratory and thus the responses to the reviewers appear justified.

RESPONSE:

We thank the reviewer for their time and consideration in reviewing this manuscript. We have responded to the issues the reviewer raised to further strengthen the manuscript.

The authors make an effort to describe the definition of new primaries but don't clarify why – is this because new primary events were discounted from PFS? One of the progressions were in the contralateral lung - was this defined as a new primary or part of PFS?

We have now rephrased and clarified this statement. Line 168 now reads: "This patient population is at very high risk for developing additional primary aerodigestive tract malignancies due to "field cancerization" effects. In order to distinguish the development of a new primary NSCLC from local or systemic disease progression, development of disease in the contralateral lung without evidence of ipsilateral or systemic recurrence were reviewed and characterized by a multidisciplinary tumor board."

In regards to the patients classified as progression, one progressed at the primary (and also metastatic disease), two progressed in the ipsilateral lung (+/- mets), and one in the contralateral lung (defined as mets). These were all defined as progression.

"Need to clarify in discussion and results that 'ORR' is 'ORR to atezolizumab' to ensure no confusion of radiation effect for the reader. This would address one of the other reviewer queries."

We have now further clarified that ORR is ORR to atezolizumab. Line 326 – has been modified to read: "We evaluated early treatment response to 2 cycles atezolizumab (before initiation of SABR)." Also, line 351 we added: "as assessed after 2 cycles of atezolizumab and before SABR" when describing responders. Line 448 in the discussion reads: Another indicator of activity was a 17% rate of early radiographic response by RECIST after two cycles of atezolizumab, prior to SABR.

The HR for the differences between DL 3 and 1 are very wide (hazard ratio: 0.156 (0.027-0.911); although statistically significant, it should be acknowledged in the results line 337 that this is exploratory and hypothesis generating.

We have modified the manuscript as suggested. Line 340 – now reads: "This analysis is hypothesis generating and the efficacy of this approach is being tested in a randomized phase III trial."

REVIEWERS' COMMENTS

Reviewer #4 (Remarks to the Author):

Thank you for addressing the comments adequately. I would recommend this for publication.